# Wireless Local Area Network Technologies as Communication Solutions for Unmanned Surface Vehicles

**DOI:** 10.3390/s22020655

**Published:** 2022-01-15

**Authors:** Andrzej Stateczny, Krzysztof Gierlowski, Michal Hoeft

**Affiliations:** 1Department of Geodesy, Gdansk University of Technology, 80-233 Gdansk, Poland; 2Faculty of Electronics, Telecommunications and Informatics, Gdansk University of Technology, 80-233 Gdansk, Poland; krzysztof.gierlowski@pg.edu.pl (K.G.); michal.hoeft@pg.edu.pl (M.H.)

**Keywords:** USV, wireless networks, WLANs, MAVLink, MAVLink-over-IP, field experiments

## Abstract

As the number of research activities and practical deployments of unmanned vehicles has shown a rapid growth, topics related to their communication with operator and external infrastructure became of high importance. As a result a trend of employing IP communication for this purpose is emerging and can be expected to bring significant advantages. However, its employment can be expected to be most effective using broadband communication technologies such as Wireless Local Area Networks (WLANs). To verify the effectiveness of such an approach in a specific case of surface unmanned vehicles, the paper includes an overview of IP-based MAVLink communication advantages and requirements, followed by a laboratory and field-experiment study of selected WLAN technologies, compared to popular narrowband communication solutions. The conclusions confirm the general applicability of IP/WLAN communication for surface unmanned vehicles, providing an overview of their advantages and pointing out deployment requirements.

## 1. Introduction

Research and development activities in a wide range of topics connected with Unmanned Vehicles (UVs) have sustained rapid growth recently, leading to practical deployments in a multitude of environments and tasks [1,2]. It is unsurprising, given that both driving elements of a classic model of high-technology systems development are present here: a technology push and a business pull. The former—availability of technical solutions allowing new solutions to be created—relates in this case to advances in both mechanical and information sciences, specially where fully or semi-autonomous vehicles are concerned. The latter—a commercial interest in such newly developed systems—is well recognized and present in multitude of fields, staring with consumer market (e.g., entertainment and competition drones), through professional systems (such as self-driving cars and various utility vehicles for general monitoring and work in difficult to access or hostile environments) and ending with customized designs for specific missions or military applications [3]. In this paper, we are going to concentrate on a specific functionality, necessary in practically all these use-cases and for all types of UVs (including fully autonomous ones)—external communication allowing UVs to be remotely monitored and controlled as needed or to offload some of their automatic functions to external infrastructure [4].

While there are many technologies (both open and proprietary) addressing the need of communication between an UV and its supporting external infrastructure or a ground station [5,6], there is an emerging trend to utilize a well-known Internet Protocol (IP) [7] for this task. The IP has been a basis of Internet for some decades now and proves well-suited to provide information transfer capabilities for a wide range of services—both these requiring bulk transfer of large amounts of information and these needing strict transmission quality parameters to be maintained to ensure a real-time response. These capabilities can also be retained in a shared communication environment, where information traffic of different services must coexist, allowing (if appropriate control mechanisms are employed) the use of a single, universal communication solution for different subsystems on-board of an UV. Moreover, an ability of the IP protocol traffic to be transported over the Internet infrastructure, allows the communication with UV to be carried from any Internet-enabled place over the globe.

The popularity of IP protocol and its supporting mechanism results in maturity of both concepts and implementations related to an information transfer in the IP-environment—both its strengths and limitations are well known and best practices for different use-cases have been developed. The above-mentioned characteristics seem to predispose the IP well for use in UVs, however its popularization in this context has been limited by a relative complexity of IP protocol stack (requiring a more advanced hardware and a correct configuration) combined with the fact, that in many usage scenarios (e.g., simple control link to an UV in direct observation of the controller) its features are not needed. In such cases, much simpler solutions, dedicated to the single task of controlling UV (and sometimes providing some rudimentary telemetry data) over direct wireless link to the controller are commonly used.

While such dedicated solutions often relay on dedicated wireless transmission technologies or narrowband radios capable of transmitting low-bandwidth data streams exchanged between serial interfaces of an UV control module and a controller, IP protocol data packets are normally carried using broadband wireless technologies. An extremely popular group of such technologies are Wireless Local Area Networks (WLANs). The most popular WLAN example is currently the Wi-Fi technology [8], whose modern implementations are capable of data exchange with speeds over 1 Gbps at ranges exceeding several hundred meters. In this situation, the ability to employ WLAN technologies for communication with UVs would both allow the easy deployment of IP communication mechanisms and a broadband data transfer capabilities. However, their suitability for particular UV-related scenarios remains to be verified.

The importance of UV external communication tasks has already been notice by standardization groups, for example European Telecommunications Standards Institute (ETSI) that proposed [9] aiming to utilize popular cellular networks for use in aerial UV communications. With the ongoing standardization works and a number of research works focused on communication link evaluation for aerial UVs already available [10,11,12,13,14], we aim to focus our research on Surface UVs—a group of lesser popularity in the consumer market (mostly oriented towards entertainment and competition drones), but of high interest for professional, commercial deployments (supporting activities such as environment monitoring, construction and maintenance of infrastructure, emergency response and even military operations) [15]. The need for research in this specific area is highlighted not only by its commercial utility but also due to the fact that Aerial and Surface UV communication links are characterized by different propagation phenomena [16,17].

This paper presents results of laboratory and field-conducted experiments aiming to verify the general suitability of IP protocol and WLAN technologies for purposes of external communication of surface unmanned vehicles. The experiments include two different, broadband WLAN technologies which our earlier research pointed out as both cost-effective and efficient in maritime communication [18,19]—the popular Wi-Fi and a proprietary Mikrotik NV2 [20]. These technologies have been used to carry an IP-encapsulated Micro Air Vehicle Link (MAVLink) protocol [21,22]—a popular UV communication protocol, supporting telemetry, configuration and direct control tasks. The above technologies have been compared to two well-known, narrowband communication solutions widely used in conjuration with MAVLink protocol—a low-cost Si1000 radio working in 433 MHz band and a long-range RFD communication module in 868 MHz band [23]. Initial experiments were conducted in laboratory environment, followed by the main measurement campaign conducted on inland waters, utilizing a HydroDron by Marine Technology [24]—an autonomous/remote-controlled multitasking water platform designed for hydrographic survey missions.

Apart from laboratory and field-grade verification of IP protocol and WLAN technologies suitability for surface UAV communication tasks obtained results allow establishing a baseline for further research concerning UV external communications due to effective lack of external interference sources during performed field-grade tests. Such a reference will highly facilitate further practical studies, especially interpretation of test results from areas where such an interference is present. Obtained measurement data includes extensive information regarding communication devices state and allows diverse analysis of traffic quality of service parameters. For the purpose of the MAVLink over IP/WLAN suitability analysis presented in the paper, it has been used to obtain an application level end-to-end round-trip time and message loss ratio of MAVLink communication.

The experiments required a design and implementation of a hardware/software test system allowing easy comparison of employed technologies. It has been realized by keeping the data processing chains of both narrowband and WLAN devices effectively identical, despite their different operational characteristics and provided physical interfaces. Moreover, the developed system utilizes a non-interfering parallel testing of all technologies in question, which further improves the precision of analysis by not requiring a comparison to be performed across separate measurement passes for different technologies. The system has been verified in laboratory conditions and deployed on a professional-grade surface UAV during a real-world deployment.

In following sections a general architecture and communication requirements of unmanned vehicles have been described (Section 2), followed by a more specific description of MAVLink protocol often employed for external UV communication tasks (Section 3). The advantages and requirements for transmitting MAVLink messages using IP protocol have also been provided and described in some detail. Section 4 is dedicated to a short description of radio-communication technologies employed in experiments, including both narrowband and WLAN solutions. Based on these characteristics, an architecture and specifics of the prepared test system have been described in Section 5. Results of experiments conducted in both laboratory and field conditions, complete with their interpretation are presented in Section 6. The final Section 7 presents conclusions regarding suitability of IP and WLAN communication for UV-related use-cases.

## 2. General Architecture and Communication Requirements of UV Control System

Despite differences in the construction of different types of UVs (e.g., aerial, surface, land, etc.) their control system usually consists of similar elements. Figure 1 show its architecture at high abstraction level required to make the architecture general. Of course, as described below, each of these general elements can be implemented using different hardware architectures, physical connections, communication protocols, logical message formats, etc.

Control elements of UV need to be supplied with the electrical *power* and in many cases its propulsion is also electric. The power system most often utilizes rechargeable batteries with appropriate control devices to allow both power delivery to a number of different subsystems and replenishment of stored energy from external or on-board sources (e.g., solar panels).

The *propulsion control system* is responsible for providing control of motors (most often electric or internal combustion) while *steering control* allows the use of elements such as flight-control surfaces, turning wheels, propellers capable of providing vectored thrust or a rudder.

An UV can obtain information about its external environment from *sensors* of many different types, starting with simple rangefinders, through GNSS receivers, video cameras and ending with 3D LiDARs and depth cameras. This information allows it to not only provide its operator with information necessary for safe and efficient control of the UV, but also required to perform autonomic control functions if such are supported.

The central control element is a *UV control board* (frequently named Flight Control System (FCS)), which is responsible for managing other control elements in a manner allowing the UV to move in a desired fashion and potentially react to unpredicted events. The control board monitors status of other UV components and external environment (using various sensors), and manages propulsion and steering control systems. It is also sometimes responsible for tasks related to a mission-specific payload, such as a photography equipment or various scientific instrumentation. The control board functions can be implemented as a single device or distributed over multiple specialized hardware elements, for example by offloading processing of data from sensors such as a LiDAR to an external computer separated from the main control unit.

The fact that the UV control system consists of many components created the need for internal *communication*. The most common technologies employed for this purpose are various types of serial point-to-point interfaces (such as a Universally Asynchronous Receiver/Transmitter (UART) or RS232 [25]), pulse-modulated signals (e.g., pulse width modulation or pulse position modulation) or more advanced, multi-point serial communication solutions such as I2C, SPI [25] and CANBUS [26].

While implementing internal communication between control system components, while not uncomplicated, is rather straightforward, the task of providing an UV with a reliable and efficient external communication capability is more difficult [27]. One of the reasons is the fact that (in contrast to internal communication between specific UV components) external data exchange requirements can vary depending on a number of factors. One of the most important is a degree of autonomy of the UV [28,29]:Remotely controlled vehicles—fully dependent on remote operator for control input;Semi-autonomous vehicles—capable of executing a detailed mission plan and supporting a number of automatic procedures allowing them to quickly react to most common, unplanned events, which do not require significant change in mission plan (e.g., avoiding an obstacle or stopping to prevent collision). As in many cases missions environment includes a high number of unpredictable elements, they are also frequently operated or supervised by remote control supplemented by automatic procedures to support the operator;Fully autonomous vehicles—capable of automatically generating their mission plan based on general objectives and automatically altering it as needed in presence of unplanned events. Fully autonomous vehicles do not require human input during normal operation.

Remotely controlled vehicles require a highly reliable, relatively low latency control communication supplemented by remote telemetry and often access to many on-board sensors (such as cameras, rangefinders, LiDARs, etc.) which allow the controller to assess the UV state and environment in real time. The bandwidth requirements of control and sensor access tend to be vastly different. The control and telemetry traffic requires relatively low bandwidth, while access to devices such as cameras requires the ability to transfer a lot of information. Due to this characteristic, it is a frequent practice to use different communication technologies for each of these tasks. While such a solution prevents a possibly dangerous contention between these traffic types, it creates new challenges, as different installation requirements need to be observed (such as antenna types and power usage) and different usage characteristics (such as possible communication range and ability to coexist with other transmitters in the same band) must be taken into account.

While semi-autonomous vehicles share most of requirements of remotely controlled vehicles, they can allow relatively less reliable and higher-latency control communication and often can be efficiently operated with limited access to on-board sensors. On the other hand, they require a reliable, but not strictly real-time data transfer functionality for managing mission plans and reconfiguration of various parameters used to configure their automated functions.

Fully autonomous vehicles in theory should be able to operate completely independently, apart from initial upload of mission description. In practice, remote control and monitoring capabilities similar to these used in case of semi-autonomous vehicles are implemented to allow supervision of their operation and manual control in emergency situations. It should also be noted, that some implementations of autonomous vehicle control systems utilize external communication links to offload specific tasks (which are computationally intensive or require access to large stores of information) to external infrastructure (e.g., multicriteria route planning based on data sets too large to store on-board [30,31]).

Apart from the control system itself of the UV, its mission-specific payload may also require a communication capability [32]. The characteristics of this communication are difficult to predict as they are fully dependent on the payload type and function [33]. In general main aspects which can be used to characterize the type of traffic are: required bandwidth [34], accepted level and variation of delay in information delivery and its reliability requirements [35,36].

When analyzing external communication characteristics one must also consider a set of general requirements not specifically tied to a particular type of UV. One of the most evident of these is a communication range a particular solution provides—in other words a distance from which we can exchange information with UV. While in most of the common cases it is equal to a transmission range of a particular communication technology (a link range), it does not need to be the case if a multihop communication system is employed. Such a system allows information to be forwarded through a series of intermediate devices before it reaches its destination.

The second characteristic is an ability to coexist with other systems utilizing the same frequency band and to maintain its efficiency even in a congested environment. The final attribute is a security level provided by the communication system. That requirement is a complex one [2], but three base aspects can be distinguished:Authenticity—an ability to confirm of the identity of the information sender (e.g., to prevent an authorized controllers from affecting an UV)—most often implemented by message signatures;Integrity—a verification that information has not been modified in transmission—possibly by use of checksums or as a part of authenticity mechanisms;Confidentiality—preventing unauthorized recipients from accessing the content of transmitted information—implemented by means of encryption mechanisms.

With the above characteristics and requirements in mind, the discussion can now be focused on methods of implementing them in practice. While different approaches to transmitting information will be presented in the next section, an introduction of protocols allowing logical organization of exchanged information is required first. While there is a number of proprietary solutions covering both data organization and transmission (such as DSMx, FASST, ACCST or AFHDS2 [37,38]), the presented analysis is concentrated on a popular open standard widely usedin the case ofsmall and middle-sized UVs—a Micro Air Vehicle Link (MAVLink) protocol [21].

## 3. Micro Air Vehicle Link Protocol

The main purpose of the MAVLink protocol is to provide a common format for information exchanged between different systems (for example UV and ground station). It utilizes the concept of messages, which are data structures serving a particular purpose (e.g., setting/reading a configuration parameter value or controlling a camera). The set of possible messages is sizable and can be extended by providing communicating devices with message definitions in XML files. This ability allows the protocol to be easily employed in support of new functionalities—related to both UV control and payloads. MAVLink also enables as much as 255 devices to use the same communication system, by delivering messages to specific devices or group of devices—created using publish-subscribe approach, where devices which subscribe to so called “topics” will receive messages published within its bounds. Both unreliable and reliable (where reception of a message must be acknowledged or it will be resend) communication modes are supported.

Due to its ability to be extended, MAVLink is currently employed for many UV-related tasks. The most popular of them have been identified and had their message formats and their exchanges necessary for specific operations standardized withing MAVLink specification as “microservices”. There are currently over 20 such microservices, supporting tasks such as defining a mission plan, manual UV control, camera operation, landing procedures, exchanging terrain description for autopilot or image transfer. It is evident that, as far as support for various high-level application and their associated data format are concerned, MAVLink is an universal protocol, fit to support most current use-cases and able to accommodate new ones. Moreover, it can be a handy tool enabling a service integration during a communication process—the ability to allow many independent services to simultaneously communicate using the same link.

Version 2 of MAVLink protocol also addresses security requirements, such as authenticity and integrity of messages, by introducing an optional signature mechanism [21]. If employed, this mechanism ensures that only devices in possession of a common secret key can exchange messages and that any unauthorized modification of said messages can be detected by communicating devices. The solution also employs timestamping, which protects from replay attacks, in which an external device attempts to record then replay legitimate messages. Such level of security should be considered acceptable, but it should be noted, that protection of confidentiality is not provided for exchanged messages. If required, it must be implemented by a transmission technology.

MAVLink messages can be transmitted using variety of means, but the most popular approach is a transmission over a serial interface, which is a popular way of implementing communication between on-board devices. To communicate with remote devices, a number of radio modules [39] providing the ability to exchange data received over serial interface over specific radio bands (serial-over-radio) are available. While this method is a very popular one, especiallyin the case ofsimple communication setups, there is also a second well-established approach, which offers some significant of architectural advantages and opens a possibility of employing a different class of communication technologies for wireless transmission—encapsulation of MAVLink messages in the Internet Protocol (IP) packets.

### MAVLink over Internet Protocol

The use of Internet protocol for transmission of MAVLink messages requires a set of more complex mechanisms to be present in communicating devices thanin the case ofa direct-link serial-over-radio service. The IP, being an ISO-OSI (ISO Open Systems Interconnection [40]) layer 3 (network) protocol, does not natively provide such a service, only an ability to deliver data packets within an environment composed of multiple devices possibly connected by many different communication links and technologies. To use this ability to carry MAVLink traffic an ISO-OSI layer 4 (transport) protocol needs to be employed over IP to encapsulate MAVLink messages. Most either Transport Control Protocol (TCP) or User Datagram Protocol (UDP) are used for this purpose (Figure 2). Of these two, the UDP is generally a preferred choice for MAVLink, because the TCP includes reliable delivery mechanism based on retransmissions which doubles similar mechanism provided by MAVLink for specific subset of its functions. Retransmission-based mechanisms provide reliability of delivery but also induce additional, difficult to predict latency, which is a drawback in many usage scenarios (e.g., real-time remote control) in the case of MAVLink these mechanisms are possible to employ as needed (depending on a particular microservice), while the use of TCP would include them in all MAVLink communications.

While implementation and configuration of IP protocol stack requires both additional resources of communicating devices and a more complex configuration as well as increases protocol overhead for each transmitted message, there are some significant advantages to such an approach.

One of the most important characteristics of IP protocol is the possibility of using it for communication over many different transmission technologies [41]. It is even possible to send its data packets over a serial connection as described before, however in most cases it would be impractical to use IP and add IP/UDP layers to protocol stack in this case. The intended transmission means for IP protocol include broadband computer network solutions such as Wireless Local Area Network (WLAN) technologies. They offer transmission bandwidth of at least a few Mbps up to a few Gbps and a range from a few hundred meters up to a number of kilometers. Such a high possible throughput allows the MAVLink protocol to take a full advantage of its service integration capabilities by being able to simultaneously deploy many different microservices over the same radio link. Moreover, MAVLink can efficiently share the link with other IP-based/Non-MAVLink services (e.g., payload related) with well-developed IP-based Quality of Service (QoS) control mechanisms ensuring intended bandwidth sharing policy or with new mechanisms designed especially for UV communication systems [42].

The IP is a network layer protocol and its main function is to ensure the delivery of data packets in a complex network environment, in particular in a network system composed of many different networks and links. Such an ability allows us to employ a number of interesting communication solutions. One such is an access network dedicated to UV monitoring and control, created by deploying a system composed of multiple access stations. That way, a stable, high quality communication conditions can be ensured over a mission area in a manner which a single communication technology would not be able to provide [5]. Figure 3 illustrates such a setup, employing a number of short-range, wireless access devices and a single long-range one, which employs a different transmission technology. Additionally, while some of these devices are connected to ground station using wired connections, other (located further away) are connected by point-to-point radio links.

Such a system can be reserved for an exclusive use of a single operator/organization or can be employed to provide UV-related communication services to external operators.

Another type of a complex IP-based system which may be of interest to UV operators is a self-organizing ad hoc network [43,44,45], where each connecting UV can serve as point of network access for other devices (Figure 4). Such dynamically created network can be of high utility in places where it is impracticable to build or maintain a static network infrastructure (e.g., over sea areas or war zones) [46].

Finally, due to this proven ability to function in complex, heterogeneous network systems [47,48], IP protocol is employed as a basis of data delivery in worldwide Internet. As a result, MAVLink-over-IP communication can be conducted directly over public Internet, opening a large number of previously impossible or impractical deployment scenarios for UV [49]. Figure 5 presents an example of such scenario, with multiple UV deployments dispersed worldwide can be monitored and controlled from many different locations.

It should also be emphasized, that MAVLink communication in all of the above systems is possible between any of participating devices, not only between a single UV and its associated ground station. Scenarios such as a ground station controlling multiple UV, multiple ground stations employed to simultaneously monitor/control different UV subsystems, a group of UVs communicating with each other or remote troubleshooting/maintenance of the vehicle by its manufacturer are very easy to implement in the IP environment.

Last but not least, an IP protocol is well suited to function in untrusted networks, where the communication should be protected with cryptographic mechanisms. Solutions such as IPSec [50], providing all previously mentioned aspects of communication security, are in use for many years and have reached a high degree of maturity. Thus, the MAVLink messages can protect even when MAVLink version 2 message signing mechanisms are not used and to a greater extend then the aforementioned solution allows.

## 4. Radio Communication Technologies

With the utility of MAVLink protocol and advantages of its encapsulation and transmission by means of IP protocol, it remains to be verified if available communication technologies are well suited for this manner of MAVLink message transmission. We have decided to focus our further attention on two main groups of technologies, which either are currently employed for UV external communication tasks or can be utilized in such a manner: direct-link radio solutions and Wireless Local Area Networks.

### 4.1. Direct-Link Radio Technologies

Solutions belonging to this group employ narrowband radio technologies for direct MAVLink communication between ground stations and UVs. They are widely used for this purpose due to their straightforward integration and configuration process. Such communication devices most often provide a serial interface for data transfer, sometimes including optional USB to serial converter and are configured by commands following at AT standard [51], which allows the same serial interface to be used for both configuration and data transfer.

Devices of this group frequently follow Listen Before Transmit (LBT) procedures [52] as defined by European Telecommunications Standards Institute (ETSI) and employ different variations of Automatic Frequency Agility (AFA) mechanisms to share the wireless transmission medium with other devices. The use of LBT allows them to disregard duty cycle restrictions strictly limiting the maximum percentage of time in which devices without LBT support are allowed to transmit on a specific channel. Listen Before Talk procedure requires a device to verify that the frequency channel it is planning to transmit on is free, which it does by listening on such channel for a specific (randomized) period of time. If transmission is detected, the device must wait for the medium to become free. If no transmission is detected, the device may transmit for a limited time. Following the transmission, it must refrain from transmitting on the channel for a specified time. LBT procedure reduces the probability of failed transmissions due to concurrent activity of many transmitters in mutual interference range (collisions) and prevents a single transmitter from occupying the channel.

Automatic Frequency Agility (AFA) mechanisms allow a device to automatically choose a frequency channel from a given set, but are less strictly defined than LBT. ETSI suggests that AFA procedures should lead to uniform spread of channel load in a given frequency band, but mechanisms implemented in many devices have additional goals, such as maximizing transmission time allowed for a device while still remaining conformant to duty cycle restrictions—it is realized by periodically switching between different channels within a specific frequency band.

Such devices can work using unlicensed frequency bands: 902–928 MHz (ITU Radio Region 2) or 433.05–434.79 MHz and 868–870 MHz (ITU Region 1). Because the bands are utilized by a number of users there are regulations limiting maximum output power, aiming to reduce interference between them. In band 868–870 MHz a high power mode can be used only if a duty cycle (percentage of time in that a device is allowed to transmit) is reduced.

The technologies do not normally include a reliable delivery mechanisms (such as retransmission-based transport protocols), but more advanced devices process data using error correcting codes allowing reconstruction of incorrectly received bits—often as many as 25%.

Technologies of this group are widely employed in simple usage scenarios (for example scenarios with uninterrupted UV visibility be the operator) due to simplicity of their deployment and, in the case ofmore advanced implementations, considerable range of relatively low-frequency, narrowband communication. However, it should be noted, that UVs often employ not a single but a number of such communication links, to provide communication separation between different UV subsystems requiring external communication, and due to limited communication resources (mostly throughput) provided by a single link. Limited resources are also a cause for joint employment of both narrowband technologies and technologies belonging to our second group of interests—Wireless Local Area Networks.

### 4.2. Wireless Local Area Network Technologies

The second group of communication technologies considered in this research are Wireless Local Area Networks. They are broadband computer network solutions designed for use mostly in popular 2.4 GHz and 5 GHz Industrial, Scientific and Medical (ISM) bands (with a new 6 GHz unlicensed band available depending on location). As computer network technologies they are intended to be used as an underlying layer for ISO-OSI network layer protocols such as IP [40] (as previously show in Figure 2) and do not to directly support transmission of data provided by a serial interface connection over which MAVLink information exchange is performedin the case ofdirect-link technologies. On the other hand, communication links (ISO-OSI layer 2) provided by WLAN devices can be used as elements of multi-element network systems (layer 3) as described in the previous section. Over such a network information is transported (layer 4) by methods appropriate for a particular application (layers 5–7)—one of which can be MAVLink information exchange or other data stream required by UV control system or its mission-specific payload. The popularity of WLAN technologies in a modern IT infrastructure is both caused by and leads to high availability their implementations in COTS (Commercial Off The Shelf) devices. The most popular of modern WLAN technologies is the Wi-Fi technology based on IEEE 802.11 standard [8], developed and extended continuously since its first publication in 1997 [53]. At first capable of providing 2 Mbps transmission speeds, it is now theoretically capable reaching almost 10 Gbps (Wi-Fi 6, based on IEEE 802.11ax standardization [54]), while its actual implementations most commonly provide up to 1.3 Gbps (Wi-Fi 5, IEEE 802.11ac [55]). Such high throughput makes it possible for data streams generated by many different services to be transmitted over a single radio link. To further facilitate this use-case, a traffic prioritization mechanisms have been developed and implemented (IEEE 802.11e [56]), allowing traffic to be divided into 4 classes—each offering different Quality of Service characteristics:Best effort—intended for bulk data transfers, with no specific real-time requirements;Video—appropriate for video streaming, providing high bandwidth, but allowing for increased delivery latency in overload conditions;Voice—making possible a low-latency, but also low-bandwidth data exchange, well suited for real-time communication and control tasks;Background—used for data transfers which should minimally impact traffic in other classes even in overload conditions.

Designed to provide means of popular, local network communication, Wi-Fi utilizes wide frequency channels (20–160 Mhz) in ISM bands and conforms with legal limitations regarding transmission power, which, depending on the particular band, range from 20 mW to 4 W of effective isotropic radiated power (EIRP). Such configuration of a radio interface tends to result in short range of communication, especially when using high efficiency modulations such as 64-QAM or 256-QAM. Because of that Wi-Fi technology includes a dynamic rate selection mechanism, which automatically chooses the best modulation and coding scheme possible to use in specific conditions, based on quality of the received signal. Such approach allows devices to retain the radio link despite decreasing signal strength, by using less efficient modulation and coding which results in reduced bandwidth. Additionally, various types of directional antennas can be employed to allow a long range communication—for example popular 120° sector antennas of 12–19 dBi gain or omnidirectional antennas of narrow vertical beam and about 6–11 dBi gain.

Popularity of Wi-Fi installations makes it necessary for the technology to incorporate advanced medium access control mechanisms, allowing multiple devices to operate over the same area and share radio resources. A combination of complex medium sensing techniques (enabling dynamic selection of frequency channel width for both coexistence and throughput improvement) and Carrier Sensing Multiple Access with Collision Avoidance (CSMA/CA) [57] protocol used for distributed medium access control make it possible for as much as tens of independent networks to operate over the same area. It should be noted, however, that CSMA/CA is a contention based protocol and its efficiency will visibly decrease with the increase of the number of competing stations operating on interfering frequency channels [58,59].

However, apart from the Wi-Fi technology there are other WLAN technologies available on the market—for example an NV2 technology developed by Mikrotik [20]. While most of its radio transmission mechanisms are based on IEEE 802.11, it employs a different the medium access protocol—a controlled access solution based on Time Division Multiple Access (TDMA) [60] principle, where transmissions in the network are scheduled by a controlling device. As a result, efficiency of medium utilization remains high even with many stations participating in the network and stability of network connection (ability to remain connected in a changing propagation environment) is also improved.

Characteristics of WLAN technologies seem to indicate, that they should be able to provide an UV with a broadband communication option, sufficient not only for MAVLink-over-IP control and telemetry flows, but also for simultaneous transmission of various bandwidth intensive data streams (such as high-quality video or sensor readings). At the same time, care should be taken to ensure appropriate WLAN communication coverage due to their potentially limited range.

## 5. Test System Design

The significant advantages of MAVLink-over-IP lead us to believe that it can be successfully employed as a robust, reliable and secure external communication solution for UVs of different types, including the ones being our special interest—semi-autonomous surface UVs. The ability to utilize either a specifically deployed (dedicated) or even public communication infrastructure composed of multiple points of radio access, combined with ability to utilize high-bandwidth technologies such as Wi-Fi or NV2 can be highly advantageous for relatively high endurance (capable of operating over large areas) and high payload (able to utilize multiple high bandwidth sensors) UVs. However, the potential barrier in its easy deployment can be a relatively short range of communication provided by a single WLAN device, compared to popular narrowband radio solutions.

To verify the practical utility of MAVLink-over-IP protocol employed over popular WLAN communication technologies, a dedicated testbed system has been designed and an experimental analysis of crucial communication parameters of such a deployment has been performed.

As the surface UV platform for the test the HydroDron by Marine Technology [24] has been employed—a multitasking surface platform designed for hydrographic survey missions. Test were conducted on inland waters, specifically on Kłodno lake where a sparse water traffic and very low utilization of required radio frequency bands allowed us to minimize a chance of unexpected external disruptions during planned experiments. The decision to seek an environment minimizing external interference made it possible for the obtained results to establish a baseline illustrating operation of communication technologies in a real-world deployment, without an unpredictable influence of external interference sources. Such results show characteristics of surface UAV field-deployment which could not be observed in laboratory conditions, but without the unpredictable external interference making their analysis difficult and uncertain. That way follow up experiments in areas where radio spectrum is congested will be able to use present results as a reference.

The HydroDron, an Unmanned Autonomous Surface Vehicle (ASV) dedicated to hydrographic survey, is a universal unit with high transportability. With dimensions: total length 4 m, total width 2 m and height in transport mode not exceeding 1 m, it can be transported by a light car trailer towed even by a passenger car and lowered into the water easily. The hydrographic equipment can be customized depending on the task to be carried out and it is possible to install the instrumentation on both bow or stern brackets making it a highly versatile platform. Moreover, the vehicle has been designed to be equipped with automatic control devices and various recording, storage and data processing solutions for navigation and hydrographic data. In cooperation with technical observation devices, it can independently perform hydrographic tasks without the need for external supervision and control while avoiding collisions with other units and objects in the surveyed area.

HydroDron is a catamaran whose floats are connected to each other by fastening beams—bow, mid and stern. A rotary, longitudinally oriented antenna gate is attached to the mid-beam. The position of the gate is controlled using an actuator—to raise it into a vertical for operation, or a horizontal one for transport of the vehicle. Attached to the aft float gauges are electric drive units containing a high-power controller (on top of the nacelle) and an electric drive motor (underwater) driving a fixed pitch propeller. In in the middle and stern sections of the floats sealed chambers are located, housing 48 V lithium-iron battery banks used to power the propulsion units. Additional 24 V lithium-iron battery banks are employed to power all other electrical and electronic equipment. In the bow area, on the upper surfaces of both floats there are two batteries of photovoltaic cells intended to recharge the batteries during operation. Each battery bank is equipped with an appropriate electronic discharging and charging controller allowing the batteries to be charged both from the photovoltaic cells and from the shore or shipboard external electrical networks. In the bow, bottom part of each float heads of single beam hydroacoustic echosounders are installed.

The antenna gantry described above, seated in the central section of the fuselage, is the main supporting element for the navigation equipment, antennas, teletransmission/streaming devices and partly for the observation equipment. The head of the LiDAR device is located centrally, with a high-resolution rotating camera located underneath. On the outer sections of the gantry antennas of a satellite navigation system receiver, data transmission equipment, vessel traffic control and video streaming devices are located. Additionally, a weather station is located between the GNSS antenna and the LiDAR device head.

The bow tie beam has a actuator-equipped, vertical boom installed in its middle zone. It performs the task of raising and lowering a dedicated mounting bracket for hydrographic instruments, such as sonar and a sound velocity probe. A low-power microwave radar, a laser rangefinder pointed in line with the craft’s motion, a similarly oriented high-resolution camera, and a motion sensor are also installed on the bow boom. In the center of the aft tie-down beam, a short vertical boom is mounted on which an actuator and an anchor windlass (used to lower the sound velocity profiler into the water to a programmed depth) are installed. A stern-facing laser rangefinder is installed on the stern beam next to the boom. The ASV is also equipped with an universal multipurpose mount for hydrographic instruments such as multibeam echosounders, high-frequency echosounders.

The buoyancy floats as well as other structural elements (antenna gate, bow, middle and stern tie beams) are made of light composite material resistant to water (including sea water) and ultraviolet radiation. Figure 6 presents the ASV HydroDron during measurement tasks in the Port of Gdynia, while a detailed model of HydroDron showing its components is depicted in Figure 7.

Such a versatile surface platform, capable of carrying and powering a significant amount of equipment, allowed the test system to be designed for parallel testing of multiple communication technologies.

To verify the utility of WLAN communication technologies and compare them with commonly used narrowband solutions, four devices have been selected for testing—two representing the traditional narrowband approach and two broadband WLAN technologies. All were chosen from a commercial off the shelf group—general purpose devices easily available on the market. They were deployed without any modifications intended to increase their efficiency in a particular use-case, apart from an appropriate configuration of their operational parameters.

The first was a narrowband, unbranded radio module operating in 433 MHz band, utilizing a Si1000 chipset [61] and a Frequency Hopping Spread Spectrum (FHSS) transmission mode. The device supports both LBT and AFA procedures. This low-cost, lightweight module provides data rates up to 250 kbps, transmit power up to 27 dBm (500 mW) and receiver sensitivity of −121 dBm. For the purpose of the tests it has been equipped with a 6 dBi antenna.

The second device used in our experiments is a long range, narrowband radio module operating in 868 MHz band which also operates using the Si1000 chipset and FHSS transmission mode with LBT and AFA support. It supports data rates up to 250 kbps, and operates with transmit power up to 30 dBm (1 W), receiver sensitivity of −121 dBm and an inbuilt high quality low-noise amplifier can provide communication at ranges up to 40 km if low bitrate settings (≤64 kbps) are configured. The device is equipped with two antenna connectors and during the tests has been equipped with 3 dBi antennas.

Both these devices represent a currently popular group of narrowband communication technologies and provide an UART serial interface for data exchange with other elements of the system, allowing for a direct physical connection to many UV control boards. The MAVLink protocol is transmitted without IP encapsulation.

With the above, popular narrowband solutions selected for reference, a decision has been made to use two different WLAN devices to create the broadband communication link. Such an approach is possible due to strict standardization of WLAN technologies allowing interoperability of a diverse range of devices, even designed by different vendors. It has been decided to employ a Routerboard Metal 52 AC [62] as an on-board communication device, due to its physical construction characteristics (small dimensions and IP55 protection), relatively low power consumption and the ease of installation, while a Routerboard mANT15s [63] (a more performant unit, featuring an integrated sector antenna) has been used for the ground station. Both devices are capable of supporting both classical, IEEE-standardized Wi-Fi 5 technology and Mikrotik proprietary NV2 protocol [20]. The devices have been configured for a standard 20 MHz channel width and 1 MIMO stream, which allows a maximum throughput of 86.7 Mbps. They are capable of using 4 times wider (80 MHz) channel width resulting in theoretical possible throughput of 433 Mbps and the mANT can additionally support 2 MIMO streams, further doubling this value (867 Mbps).

The mANT includes a conveniently integrated 120° sector antenna providing a 15 dBi gain. This element, combined with a high maximum bandwidth, high sensitivity (−96 dBm signal sufficient to maintain a 6 Mbps link) and maximum transmission power of 31 dBm (1.26 W) makes it a good solution for a ground station communication device. Two such devices configured have been deployed to simultaneously support both Wi-Fi and NV2 communication links, created on independent frequency channels in the 5 GHz ISM band.

The on-board device (Metal AC 52) has been equipped with a 11 dBi omnidirectional antenna and configured to connect to the ground station mANT device. On-board device supports only one MIMO stream and has a bit lower sensitivity (−93 dBm for 6 Mbps link). Its maximum transmission power is the same asin the case ofmANT (31 dBm, 1.26 W). The HydroDron has been equipped with two such devices, one configured for Wi-Fi and one for NV2 operation, each connecting to its respective mANT at ground station.

All the above devices belong to a relatively low price range and can be acquired for under 150 USD each.

For the purposes of the experiments, the actual transmission power of both ground station and on-board device has been limited to conform with local UE regulations for on-shore use [64,65].

In order to minimize the time necessary for experiment and to provide easily comparable results, a test setup allowing all four communication links (low-cost Si1000 at 433 MHz, long-range Si1000 at 868 MHz, Wi-Fi 5 at 5 GHz and NV2 at 5 GHz) to be tested simultaneously have been designed and implemented. For this purpose, the system includes four separate radio links connecting two industrial computers—one at ground station and one on-board of the HydroDron (Figure 8). These computers serve as sources and destinations of MAVLink messages: ground station computer emulates a ground station using MAVLink protocol, while on-board computer emulates control system of an UV.

Narrowband radio devices provide UART data interfaces, so they are connected with an industrial computers emulating a ground station and an UV control system with USB-UART converters. Such setup allows computers to exchange MAVLink messages using USB-connected serial interfaces.

WLAN devices are connected to the computers with 1 Gbps Ethernet interfaces providing both network connectivity and power (using power over Ethernet technology). At the ground station WLAN devices are directly connected to the industrial computer, as popular ground station hardware frequently provides an Ethernet interface. On the other hand, popular UV control system hardware solutions tend to provide only serial interfaces, so at the UV, each WLAN link is connected to a single-board computer (Raspberry Pi 3B [66]) performing MAVLink-over-IP to MAVLink-over-serial (UART) conversion. UART interfaces of these protocol conversion devices are connected to the industrial computer emulating UV control system by USB-connected serial interfaces. Such an architecture, making the uniform use of USB-connected communication interfaces (both serial and Ethernet), allows an easy comparison between narrowband and WLAN technologies.

## 6. Experiments and Results

The system setup described in the previous section has been designed to allow a comparison of traditional narrowband radio communication links with the IP-based approach and its deployment using WLAN technologies. With both significant advantages to IP/WLAN approach and potential issues regarding communication range of these technologies, the question of its viability in a real-world surface UV operation is an important one. Because of that, experiments described in this paper were aimed at obtaining key quality characteristics of IP/WLAN MAVLink communication and their analysis in comparison with solutions generally recognized as proved in UV external communication tasks.

For this purpose, we have employed standard Quality of Service (QoS) indicators referring to requirements mentioned in description of autonomous vehicle communication requirements: message loss ratio (MLR)—a percentage of messages lost in transmission, communication delay measured as round trip time (RTT)—a time required to receive a response for a request sent from a ground station to an UV and available throughput of a communication link. Due to our main focus being on verification of MAVLink protocol operation, of these three important indicators we concentrated on message loss and transmission delay as being the most important for low-bandwidth MAVLink data stream. The available bandwidth is of lesser importance, as MAVLink control and telemetry links are currently successfully carried by narrowband links of less than 250 kbps, while the minimum data rate of Wi-Fi and NV2 is 1 Mbps. However an assessment of this parameter is also included to verify if further experiments concentrated on service integration using WLAN communication links are warranted.

During the experiments PING microservice [67] has been used to measure round trip time and message loss ratio. PING messages were send every 500 ms from the ground station to the UV control system emulator, which responded immediately with a similar message transmitted back to ground station. As MAVLink PING messages contain a sending time stamp and a sequence number they allowed the calculation of the RTT and MLR values. The message size was 112 bits, which resulted in a MAVLink data stream of 480 bps. In case of MAVLink-over-IP scenarios the preferred UDP encapsulation has been employed.

At the same time, a number of radio link parameters (e.g., signal strength) were collected from all deployed radio devices. Moreover, a GPS receiver has been used to obtain time synchronization, as well as HydroDron location and its movement details.

### 6.1. Laboratory Experiments

To verify the operation of the described test system, first experiments in which it was employed were conducted in laboratory conditions. Apart from verifying the basic correctness of its operation, these experiments confirmed independence of radio frequency channels chosen for different communication technologies and provided performance data for a system operating in controlled propagation conditions. Such information enabled us to create a baseline system profile used in subsequent experiments to quickly verify the viability of measurements obtained in real-world conditions.

Another aim of the laboratory experiments was to verify the impact of the TDMA Period Size (TDMA PS) parameter on the received QoS results, and select its value for the real-word tests. In general, the parameter indicates how often a station will have an opportunity to transmit a data packet and it is an important configuration element of NV2 TDMA-based medium access mechanisms. Lower values can be expected to provide faster network response, but may lead to inefficiency in a network with a high number of participating devices or under a high traffic load.

In Figure 9, a dependency of Round-Trip Time (RTT) of MAVLink messages on different values of TDMA Period Size has been presented. Additionally, RTT results for the Wi-Fi link (which uses a contention-based CSMA/CA protocol and does not support this parameter) are included for comparison. The test has been performed in near-perfect propagation conditions, with received signal strength at −38 dBm and no interfering signals.

The results visible in Figure 9 clearly show dependency between discussed parameter and MAVLink message RTT. With the experiment being conducted in otherwise unloaded network, it can be seen that TDMA medium access control will generally induce the message delay lower than two times the TDMA PS value. It is in keeping with expectations, as the message can be expected to wait up to TDMA PS for the opportunity to be transmitted in each direction. Automatic TDMA PS selection also seems to follow logical rules, choosing minimal Period Size when the network traffic is low.

Another expected but interesting observation is the low RTT value for a standard, contention-based Wi-Fi access. Following CSMA/CA rules, if a station want to transmit and the medium is free for a short period (without traffic class prioritization in use—34 µs) it can commence the transmission. It means that over an unloaded channel, CSMA/CA access can be expected to be faster than TDMA approach used in NV2.

These results lead to a decision of choosing a 2 ms TDMA PS value for further experiments as it minimizes the important RTT value while still maintaining a stable QoS level in a case of a UV-dedicated network (where we can expect a limited number of participating devices).

### 6.2. Field Deployment Experiments

Having completed the preparatory experiments in laboratory conditions, a test deployment of HydroDron has been prepared. It was intended to verify and compare classical narrowband technologies and WLAN solutions in real-world operational conditions of a semi-autonomous surface UV. The tests included a short (a few hundred meters) and medium range (up to 2.5 km) communication scenario. No attempt was made to discover a maximum possible communication range offered by technologies in question, as the aim was to test the viability of WLAN-based communication in a standard usage scenario. Previous studies show that, with appropriate configuration, it is generally possible to maintain a communication link using WLAN technologies as long as direct line of sight between antennas of a ground station and a surface UV is present—which can be as long as 14–16 km [18,68,69].

The deployment consisted of the previously described ground station with its antennas installed at about 4 m above the water level on a southern shore of Kłodno lake (Figure 10) and the equipment on-board of HydroDron emulating an UV control system using MAVLink protocol. The transmission power of all devices has been limited to strictly conform with local and UE regulations for on-shore use, as indicated in system description. Following the best practice described earlier, ground station communication devices employed 120° sector antennas. For the purpose of experiment, the location of the ground station antennas has been selected in a way which caused the leftmost 40° of the sector to be partially obscured by foliage located in its direct neighborhood. The vegetation introduced an additional, variable signal attenuation of a measured value between 3 and 8 dB [70]. Such a setup allowed us to assess risks related to suboptimal deployment of ground station antennas.

As shown in Figure 10, the activity of the HydroDron during the experiment included both high-speed movement and low-speed maneuvering. At all times, the system maintained the measured exchange of MAVLink messages as described before, using all narrowband and WLAN communication technologies in parallel, using independent frequency channels.

Numeric results of the experiment have been summarized in Table 1—they include a maximum communication distance, an average MAVLink message loss ratio (MLR) and an average communication round trip time.

Detailed analysis and discussion of these results follows, however it is evident that the low-cost Si1000 radio module operating at 433 MHz offers a significantly lower performance that any of the other devices. It was the only device which was not able to provide a coverage for the HydroDron over the entire test track due to insufficient maximum communication range. Its inability to do so resulted in loss of 90% MAVLink messages transmitted during the test, which lead to a decision to remove the device from further, detailed analysis and comparison as a clearly underperforming solution. The results show, that it can be employed in good propagation conditions for ranges up to 300 m (average MLR of 0.33), which in case of surface UVs we need to consider to be a relatively short range. In this situation, further analysis will concentrate on one narrowband technology (RFD in 868 MHz band) and two WLAN solutions: IEEE 802.11-compliant Wi-Fi 5 and proprietary NV2, both in 5 GHz ISM band.

As the received signal strength is one of the most important physical parameters influencing operation of radio technologies, in Figure 11 we have presented a map visualizing its values in data points marked at fixed intervals of 15 s. The information has been obtained from Wi-Fi, NV2 and RFD on-board devices. It should be noted, that the devices report a value of this parameter only when they consider the communication link to be active. If a particular technology has been unable to provide communication between UV and ground station no measurements have been recorded.

It can be seen, thatin the case of WLAN technologies the measurements on the leftmost leg of the track plot indicate a lower signal strength then points in similar distance located more to the right, confirming the negative impact of obstacles located between communicating devices. Moreover, lack of measurements in Wi-Fi plot in this area indicates a loss of connectivity for this technology in the case ofmore resilient NV2 the same conditions were sufficient to maintain a link. When analyzing the above results we need to remember, thatin the case of WLAN technologies link activation requires a number of higher level procedures to be completed, such as searching for available networks, network selection based on number of criteria (e.g., network name, supported security mechanisms), authentication, association and dissemination of cryptographic keys [71]. Each of the procedures requires a successful exchange of messages between communicating devices. As a result, it is much more difficult to setup a link in difficult conditions thenin the case ofnarrowband technologies which generally utilize preconfigured parameters and do not perform multi-message parameter negotiation during the process. In this situation, the ability of NV2 to keep the existing link operating even in difficult propagation conditions is especially important.

The long-range, narrowband RFD technology maintains the communication capability during the entire experiment, as can be expected from technology advertised as suitable for communication at over 30 km ranges. It can also be seen, that RFD received signal strength level is about 10 dBm higher thanin the case of WLAN technologies, as a result of differences in signal propagation in 868 MHz (RFD) and 5 GHz (WLAN) frequency bands. The foliage induced attenuation of signal in 868 MHz band is visible, but to a lesser degree thanin the case offrequencies exceeding 5 GHz.

The above results indicate that COTS WLAN devices can be provide a coverage for short and medium ranges. They performed significantly better than the low-cost 433 MHz solution and with the sector antenna and the unobscured transmission path between communicating devices they provide connectivity as stable as a high-quality RFD device. However, if the path is obstructed the classic Wi-Fi solution experiences losses of connectivity (which underscores the need of careful system design in its case), but TDMA-based NV2 technology remains stable even then.

Having verified the basic connectivity, we can proceed with further analysis of the communication quality. While the wireless link is operable, the parameter which impacts the reliability of MAVLink operation the most is the message loss ratio. The impact of a single lost message can be expected to be limited, because for microservices which require a reliable message delivery (e.g., configuration changes or file transfer) the MAVLink protocol implements it by means of retransmissions. Other microservices generate a continuous stream of messages (e.g., telemetry) and a loss of a single one will disrupt their operation only momentarily (e.g., lack of a measurement for a specific time point). Repeated losses of MAVLink messages however, can lead to a serious service degradation, inducing delays in control loop, inability to reconfigure an UV, partial telemetry information or intermittent unresponsiveness of specific controls.

Figure 12 presents MLR measurements for communication technologies under test recorded as mean values over intervals of 15 s.

In case of RFD we are observing a continuous, low loss ratio (under 10%) with only sporadic, isolated locations experiencing higher MLR values. This result is expected from a high-quality, long-range, narrowband communication device operating at short range (not exceeding 2.3 km, while the maximum advertised range of employed devices exceeds 30 km) in the environment lacking external interfering transmissions.

Wi-Fi devices show the lowest MLR in good propagation conditions, however it raises very quickly with their degradation. Moreover, a growing MLR level is a sure indication of an imminent loss of Wi-Fi communication link.

The NV2 shows a fluctuating level of MLR and the losses are a little higher thanin the case of Wi-Fi in good conditions. However, in degrading conditions the NV2 link remains operational and loss ratio grows significantly slower. As a result NV2 provides a much more stable service (despite fluctuations) and high resistance to unexpected events, such as UV leaving the expected area of operations (antenna sector) or weather changes.

Figure 13 and Figure 14 provide further, statistical study of MLR measurements. Figure 13 shows an empirical cumulative distribution function (ECDF) of MLR values obtained during experiment, while Figure 14 presents their histogram, to provide a different representation of essentially the same information.

Both figures confirm the conclusions derived from results presented on maps. It is evident from the ECDF plot thatin the case ofthe narrowband RFD radio over 80% of measurements show message loss ratio below 10%—the same can be observed on the histogram, where the green values are concentrated in the same range. On the other hand, the ECDF chart shows that even in very good conditions some loss of messages (about 4%) can be expected.

In case of Wi-Fi and NV2 technologies, it can be seen that there is a considerable probability of lossless transmission—the starting value (for 0 MLR) of ECDF is 0.39 and 0.23, respectively, and a distinct peaks are visible in the histogram. These results confirm an efficient operation of both of these solutions during a considerable part of the experiment. However, in the case of Wi-Fi the ECDF plot grows relatively slowly until there is an instant increase at MLR value of 1. Such results indicate that the quality of Wi-Fi communication can be expected to change between radically different values of MLR and confirms that for some part of the test the Wi-Fi link remained inoperable. The histogram presents this observation with an evident peak at MLR of 1.

In case of NV2 the growth of ECDF is more steep, which show that this technology can be expected to react to degradation of conditions more gradually. The NV2 ECDF values are higher than Wi-Fi ones starting from about 0.3 MLR, which shows that in other than very favorable conditions the NV2 provided a better reliability of MAVLink message delivery.

While the above statistical results seem to confirm the conclusions based on MLR map analysis, Figure 15 will allow us to illustrate possible connection between the MLR values and the received signal strength level. Apart from measurement points obtained during the experiment, we have also included trend-lines plotted using linear regression.

The results confirms that the low-throughput RFD maintains a low level of message loss and its dependency on signal strength observed at ranges covered by the test is negligible (as shown be effectively horizontal trend-line) in the case ofthis technology the signal strength recorded during experiment (between −93 dBm and −60 dBm) is much higher than the advertised receiver sensitivity of −121 dBm. Such high sensitivity explains the lack of impact of recorded signal strength changes on MLR—for the whole experiment the device operated at what should be considered a short range. Some measurement points are scattered above the trend-line (indicating higher levels of MLR) but they are relatively few.

In contrast, Wi-Fi link reacts with quickly increasing MLR to a reduction of received signal strength (steeply sloped trend-line). Additionally, measurement points are strongly scattered over the whole possible MLR range for signal strengths below −76 dBm, indicating unpredictable loss rate in other than very good propagation conditions. There is also a sizable grouping of measurement point at MLR value of 1, showing periods of communication loss.

In case of NV2, a degradation of link quality caused by reduction of signal strength is visibly slower (as indicated by a mediocre slope of the trend-line). The scattering is also visible, but measurement points tend to group below MLR value of 0.55 with almost no measurements at MLR value of 1, indicating that the link remained active and operating with adequate quality during entire experiment. Moreover, MAVLink messages have been delivered by NV2 at a considerably lower minimum signal strength (about 3 dBm) then it has been possiblein the case of Wi-Fi, as indicated by starting points of respective trend lines.

It can be seen, that all the technologies provide small RTT values (Figure 16, however there are observable differences between their performance. The narrowband RFD radio consistently maintains a relatively higher RTT value (about 0.3 s) then WLAN technologies. In this case the higher RTT is a direct result of its low throughput which makes the time needed to transmit even a single MAVLink message observably longer. The consistency of RFD results is also easily explained, as the technology does not include any retransmission mechanisms which could cause a delayed delivery of a message previously lost in transmission. The LBT mechanisms, which could also induce a variation in RTT values did not do so, confirming the lack of interference from outside signal sources.

Due to their high throughput, both Wi-Fi and NV2 provide much lower RTT (below 0.1 s), which is clearly visible in almost all point of the UV track. Due to low traffic load and contention based access Wi-Fi is able to provide even lower RTT then NV2 (as can be expected based on results of laboratory experiments). The Wi-Fi also shows relatively minor fluctuations of RTT and they are located in direct neighborhood of areas where the technology was not able to maintain connectivity. Such location seems to confirm the observation that environment change capable of disrupting the operation of lightly loaded Wi-Fi link generally leads to its disconnection, despite its retransmission mechanisms. NV2 on the other hand, while providing only slightly higher RTT in good conditions has been able to maintain the link over the complete test track. The places with higher RTT values indicate locations where NV2 retransmission mechanisms were required to successfully deliver a MAVLink message.

In this situation, the NV2 technology seems to be an attractive choice for a broadband communication technology to be used to support a MAVLink-based, surface UV communication. Its ability to maintain a link in more challenging conditions and low latency of communication correspond with requirements for such an use case.

The above conclusions are confirmed by statistical analysis of measurements, presented in Figure 17 (ECDF of RTT values) and Figure 18 (histogram of the same dataset).

Both the ECDF plot and the histogram for RFD radio indicate that 98% of RTT measurements fall between 0.23 and 0.32 s, indicating that transmission delay of delivered MAVLink messages remains largely unchanged, for reasons indicated above.

In case of WLAN technologies, Figure 17 indicates that about 85% of RTT measurements (received responses) show round trip communication delay to be under 0.05 s, with Wi-Fi holding a slight advantage over NV2. At the same time NV2 managed to fit almost 98% of its RTT measurement within 0.3 s boundary.

The association between RTT measurements and received signal strength for the three analyzed technologies is presented in Figure 19.

Measurement points for RFD are located very closely to the trend line, which is almost horizontal confirming minimal impact of received signal strength on round trip time of delivered messages.

Most of measurement points for Wi-Fi are also concentrated, but a number of them is scattered, indicating its sometimes unpredictable behavior for signal strength below −75 dBm. The Wi-Fi trend line is close to horizontal, showing effectively no direct relation between falling signal strength and increasing RTT, because its effect can be observed in increased percentage of lost messages (as shown in Figure 19).

In case of NV2, measurement points are also concentrated and while more of them is scattered, they also are located generally closer to the trend line thanin the case of Wi-Fi. The trend line itself is slightly sloped indicating an existing relation between the received signal strength and RTT values. This observation is consistent with the analysis presented above—in case of NV2 a much higher percentage of MAVLink messages is successfully delivered even in difficult conditions, but in part due to operation of retransmission mechanisms inducing additional delay.

The above statistical results further confirm the positive opinion of NV2 utility in providing external communication for surface UVs employing the MAVLink protocol. With both relatively low message loss ratio and a communication delay much smaller than the narrowband RDF solution, it can be effectively employed for short and medium-range MAVLink communication.

With the analysis of key parameters describing the quality of MAVLink data transmission indicating that WLAN technologies can be effectively used for this purpose, a further study of their broadband communication capability has been performed. Figure 20 presents an assessment of an available transmission throughput remaining while maintaining the test MAVLink data stream. The assessment takes into account both a current data rate reported by communication devices and an expected data loss, based on previously analyzed MLR parameter.

In case of the narrowband RDF link, the data rate is preconfigured at default value of 200 kbps and constant. That it is enough to carry additional MAVLink traffic streams for microservices of the basic MAVLink microservice set (which generally require a traffic stream below 50 kbps). However, it is not sufficient for any resource intensive transmission (such as video or real-time 2D/3D sensor data) so any service integration over the link must be performed with great caution. It is advisable to reserve such a link strictly for critical, low-bandwidth services.

On the other hand, results for broadband WLAN technologies show that both Wi-Fi and NV2 still had a significant amount of free communication resources available. Both of these solutions have been configured to use 20 MHz channel and a single MIMO stream, resulting in a theoretical maximum data rate of 86.7 Mbps. However, they also both employ a dynamic rate selection mechanism. This mechanism dynamically changes their modulation and coding scheme to fit current communication conditions, so the available throughput value changed during the test in the case of Wi-Fi, the available throughput has been strongly dependent on conditions, but even in moments directly preceding losses of communication link it was at least 3 Mbps. In good conditions as much as 40 Mbps of available bandwidths remained for other services to use.

As could be expected based on previous analysis, the much more resistant to adverse propagation conditions NV2 technology provided a much more stable level of available bandwidth, with a vast majority of measurements showing over 35 Mbps and only few falling below a 10 Mbps boundary.

Such results confirm, that WLAN link can be used as multi-service communication solution. Moreover, the popularity of IP protocol results in availability of many service implementations supporting this mode of communication. However, because of dynamic changes of the data rate of the link, a care should be taken to deploy appropriate traffic prioritization mechanisms to prevent traffic streams of critical services from being disrupted by streams of secondary importancein the case ofa sudden data rate reduction.

## 7. Conclusions

In the paper the utility of the popular Internet protocol for surface unmanned vehicle related communication tasks have been presented, specifically in combination with a well known MAVLink protocol. For this purpose general communication requirements of different types of unmanned vehicles have described as well as the functionality, characteristics and requirements of the MAVLink. It is clear that its ability to be carried using both serial interfaces popular in UVs internal communication systems and by encapsulating its messages in IP packets (allowing transmission over complex network systems) makes it an universal solution fit for many use-cases. Moreover, the fact that MAVLink lets its functionality to be extended by implementing new microservices allows it to be employed for an ever growing number of procedures and allows it to integrate various UV control and monitoring tasks. However, the use of IP protocol brings its own requirements, most importantly its protocol stack is a relatively complex one compared to popular, simple MAVLink-over-serial connection communication options. Because of these requirements, a description of Wireless Local Area Network technologies (a very popular group of technologies widely used for IP communication) has been presented. The analysis indicates that WLAN technologies could be a good, inexpensive choice for deploying MAVLink-over-IP scenarios. However, due to their vastly different characteristics compared to simple, narrowband devices, the suitability of WLAN technologies for surface UV operations had to be verified. The included analysis confirmed it in general principle, but also pointed out possible problems—especially related to a limited transmission range of WLAN communication and its vulnerability to physical obstacles. Further verification of the WLAN functional characteristics has been conducted in a real-world testbed environment.

For that purpose a test setup consisting of four selected communication technologies (two narrowband and two WLANs) has been designed and installed on board of a semi-autonomous surface UV, the HydroDron. The test system features an uniformed data processing paths for all employed technologies and utilizes a non-interfering parallel testing technique, facilitating comparison of results by ensuring that measurements are taken in the same exact conditions (and moment) for all of them. The UV has been subsequently deployed for a cruise over the Kłodno lake, allowing the communication solutions to be tested in their real, intended operational conditions (including some commonly encountered signal propagation obstacles), at ranges up to over 2 km. The test site has been verified to be effectively free of interference from external radio systems utilizing the same frequency bands, which allows to use the results to establish a baseline for further research concerning UV external communications. Such a reference results will facilitate further field studies conducted in areas where such an interference is present.

The tests confirmed the suitability of well-known narrowband solutions, stressing the significant performance differences within this group and the importance of choosing high-quality devices over low-cost ones. Additionally they indicated, that the limited throughput offered by such solutions, while suitable for today’s common scenarios of their employment, will most probably be insufficient to support service integration and communication needs of new UV functionalities.

The results of WLAN technology testing confirmed both their suitability for surface UV communication tasks and the necessity to take their specific usage characteristics into account when deploying them in such a role. While both of them, configured following on-shore usage legal rules, offered communication range sufficient to provide coverage of the entire lake and throughput of a few tens of Mbps, the popular Wi-Fi 5 technology proven to be susceptible to any obstructions in the signal path. The technology remained highly efficient in good signal propagation conditions, with almost no MAVLink message loss and the value of round trip transmission delay not exceeding 0.05 s—about 1/6 of the result of narrowband RDF radio. Such a good performance makes it potentially suitable even for a low-latency, real-time remote control, such as offloading some control logic to infrastructure outside of UV. However, even a medium density foliage located between stations caused throughput reduction from 30 to 40 Mbps to about 3 Mbps and subsequent, intermittent losses of communication link. Due to this lack of reliability, Wi-Fi can be considered a good solution for non-critical UV communication procedures (e.g., data transfer, configuration changes). If it needs to be employed for critical remote control procedures, a care should be taken to properly design the communication system (e.g., by employing multiple ground station communication devices covering the area of operations) to avoid the described link instability issues.

At the same time, the NV2 technology (based on the same physical layer mechanisms, but employing TDMA instead of contention-based medium access) retained stability in all test conditions, always providing at least 10 Mbps of throughput with the majority of measurements showing over 35 Mbps. It should be noted, that throughput of both NV2 and Wi-Fi can be further extended by employing a link with more parallel MIMO streams. Moreover, in the case ofthe stable NV2, more throughput can be obtained by using wider frequency channels in the case of Wi-Fi such an operation could lead to even greater service degradation in poor propagation conditions. The tests of NV2 in good conditions show MAVLink message loss and their round trip time closely comparable to Wi-Fi, but without the quick degradation of these parameters in more challenging communication environments. As a result, this technology can potentially be employed to provide MAVLink-over-IP communication suitable for low-latency control, autonomous control functions implemented remotely and service integration over a single wireless link.

The obtained results indicate, that while the tested WLAN technologies can both be deployed as wireless links supporting MAVLink-over-IP communication, the TDMA-based NV2 seems better suited for this role. Its ease of deployment, good quality of service parameters, stability in varied conditions and high throughput make it a good solution for the considered (short and medium) communication ranges. As such it can become an important technological enabler, allowing a full potential of IP-based MAVLink communication to be realized using inexpensive COTS hardware.

## Figures and Tables

**Figure 1 sensors-22-00655-f001:**
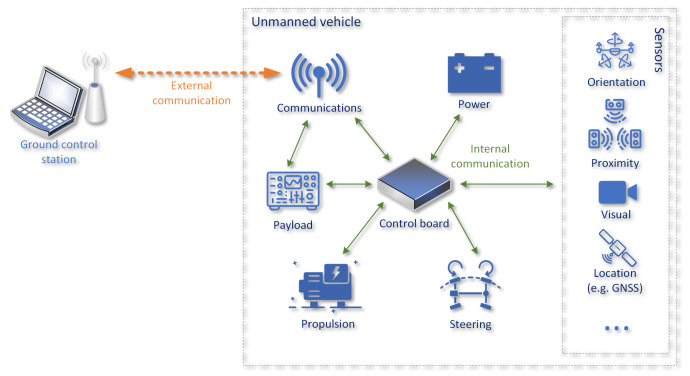
General architecture of UV control system.

**Figure 2 sensors-22-00655-f002:**
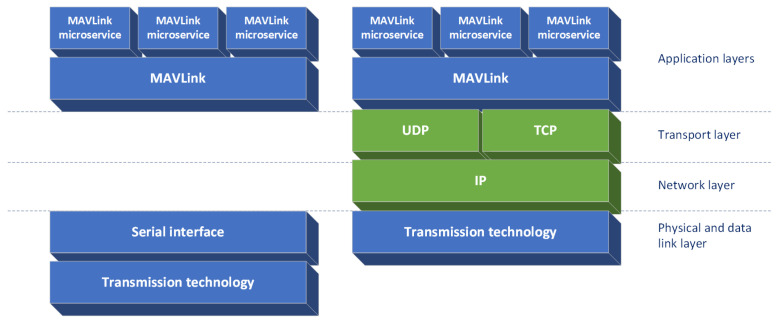
MAVLink-over-serial and MAVLink-over-IP protocol stacks.

**Figure 3 sensors-22-00655-f003:**
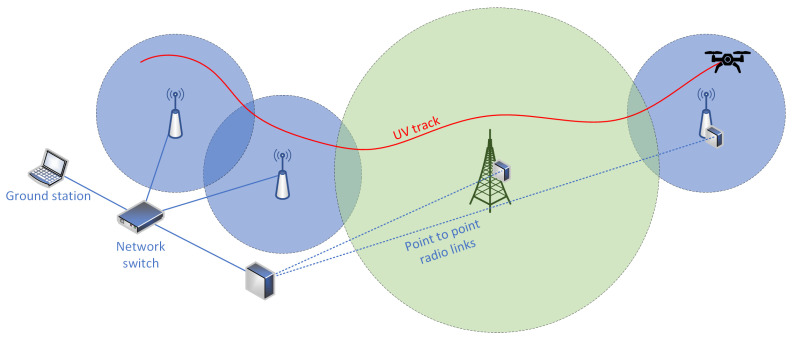
Employment of multi-technology (heterogeneous) access network for UV communication.

**Figure 4 sensors-22-00655-f004:**
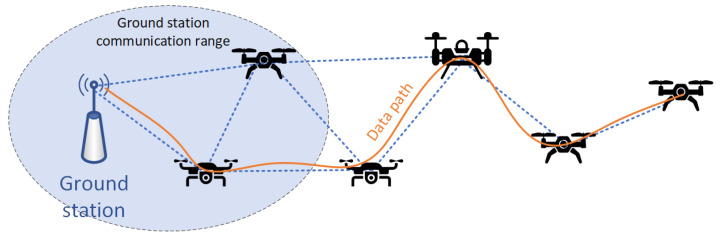
A self-organizing, multihop network utilizing UV-based nodes.

**Figure 5 sensors-22-00655-f005:**
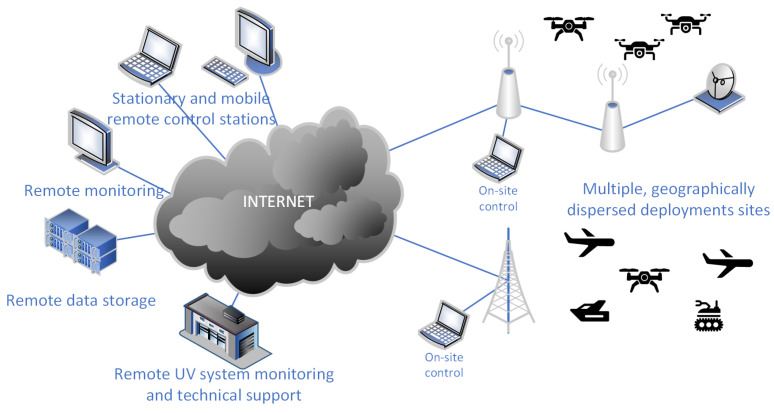
Use of Internet for remote UV monitoring, control, data acquisition and maintenance.

**Figure 6 sensors-22-00655-f006:**
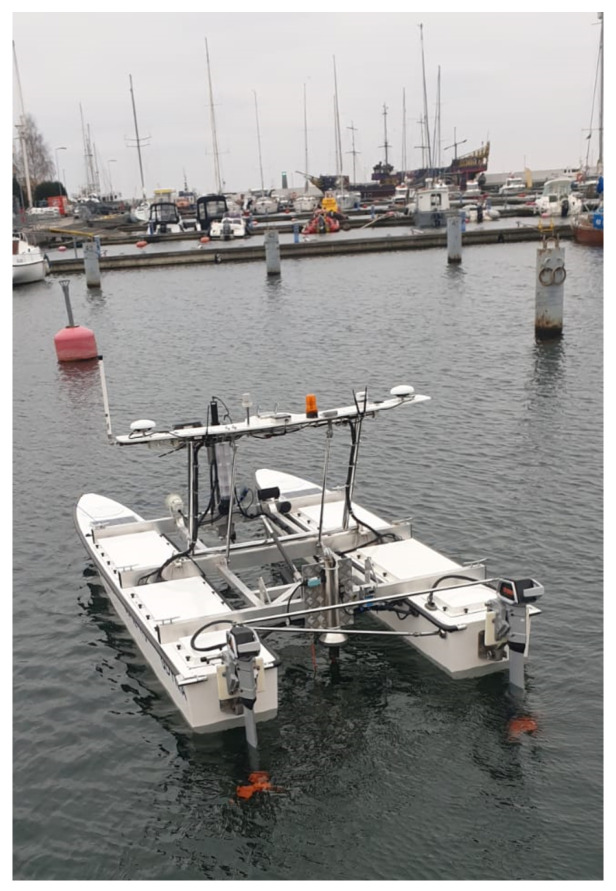
HydroDron during measurement tasks in the Port of Gdynia.

**Figure 7 sensors-22-00655-f007:**
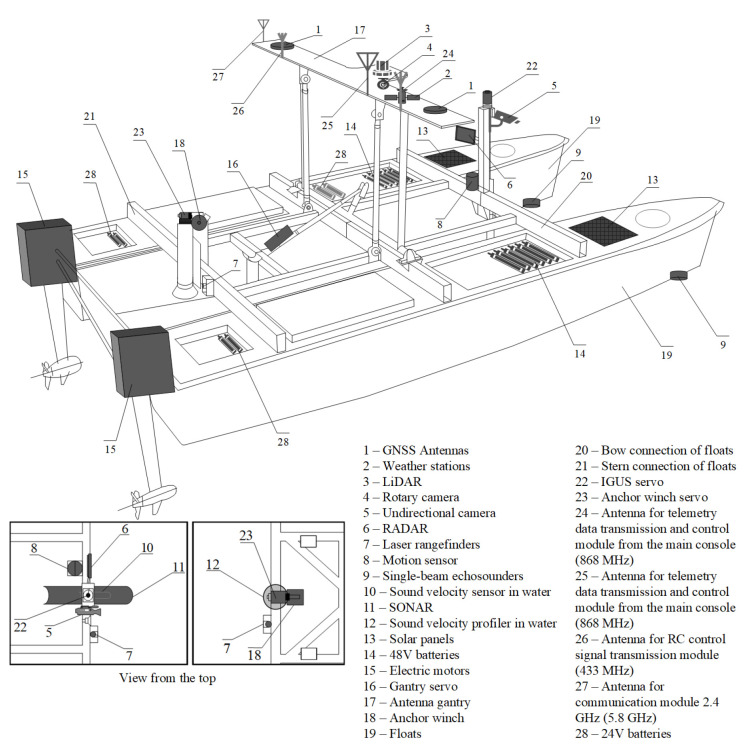
HydroDron model and its components.

**Figure 8 sensors-22-00655-f008:**
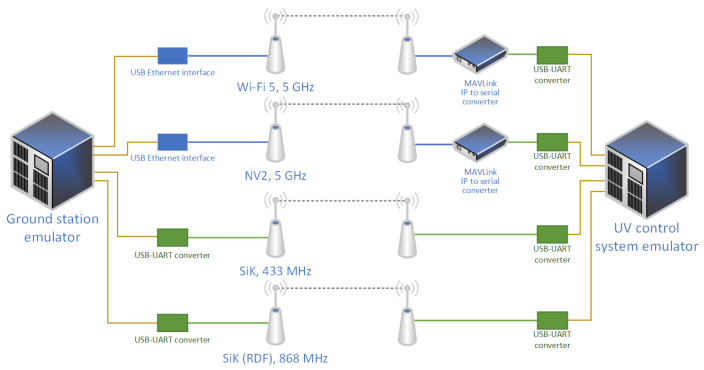
Architecture of the test system. Connections: orange—USB, green—UART (serial), blue—1 Gbps Ethernet, dotted—wireless.

**Figure 9 sensors-22-00655-f009:**
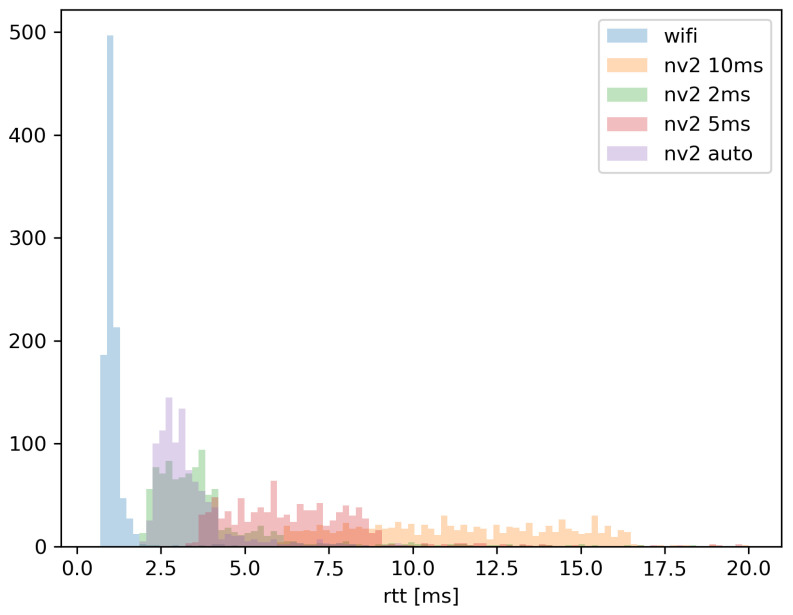
Number of MAVLink messages received with a specific RTT value for different TDMA Period Size settings.

**Figure 10 sensors-22-00655-f010:**
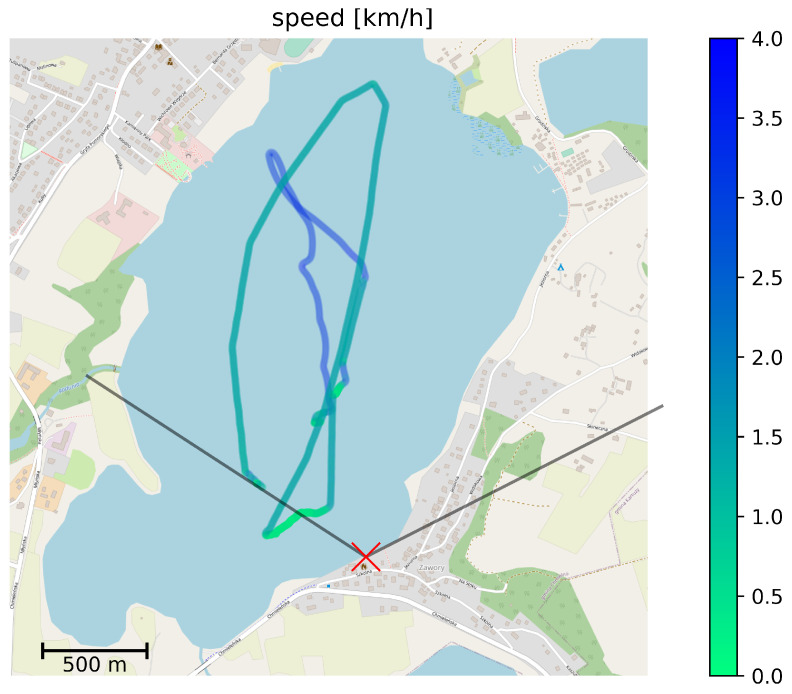
Ground station location (marked by red X), its 3 dB antenna sector boundary and HydroDron movement speed (km/h) during the test deployment.

**Figure 11 sensors-22-00655-f011:**
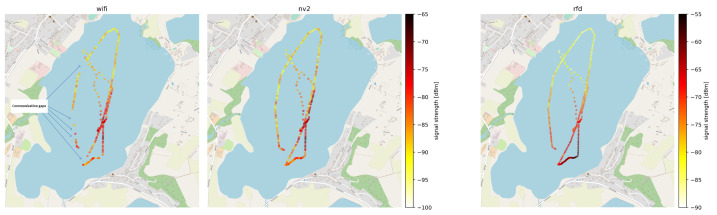
Received signal strength for Wi-Fi, NV2 and RFD devices during HydroDron test deployment.

**Figure 12 sensors-22-00655-f012:**
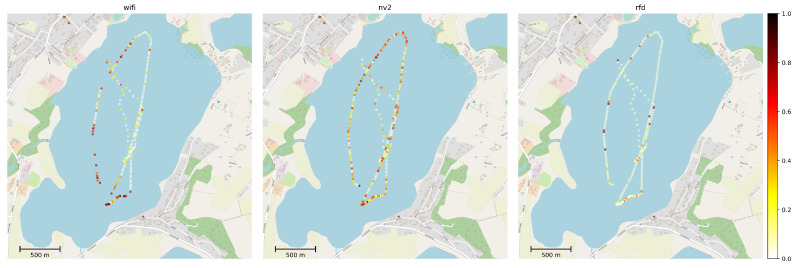
MAVLink message loss ratio measurements for Wi-Fi, NV2 and RFD devices.

**Figure 13 sensors-22-00655-f013:**
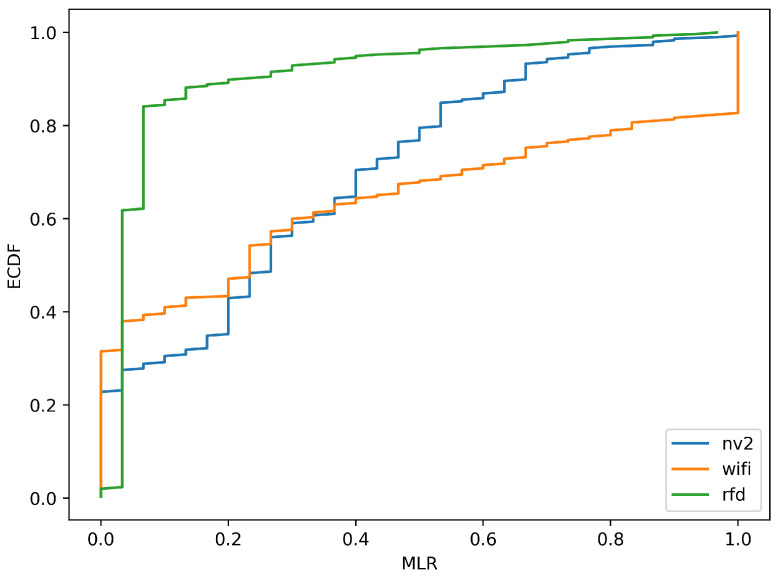
ECDF plot of message loss ratio results.

**Figure 14 sensors-22-00655-f014:**
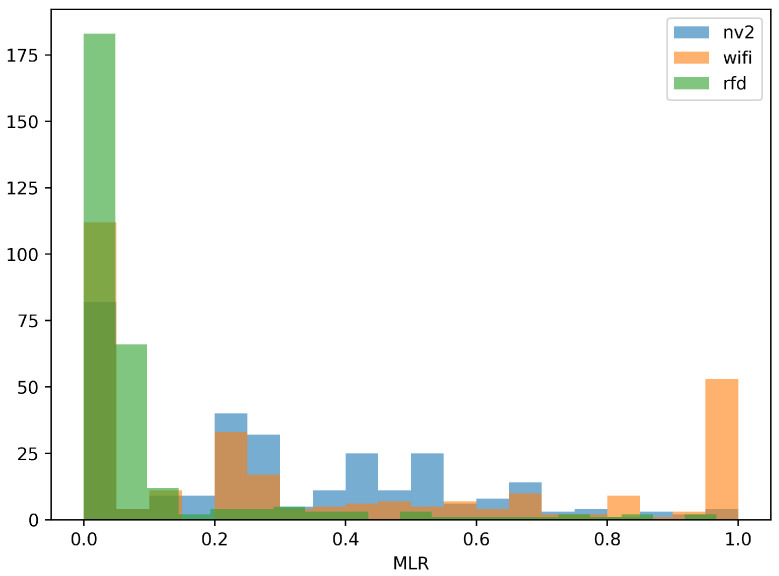
Histogram of message loss ratio results.

**Figure 15 sensors-22-00655-f015:**
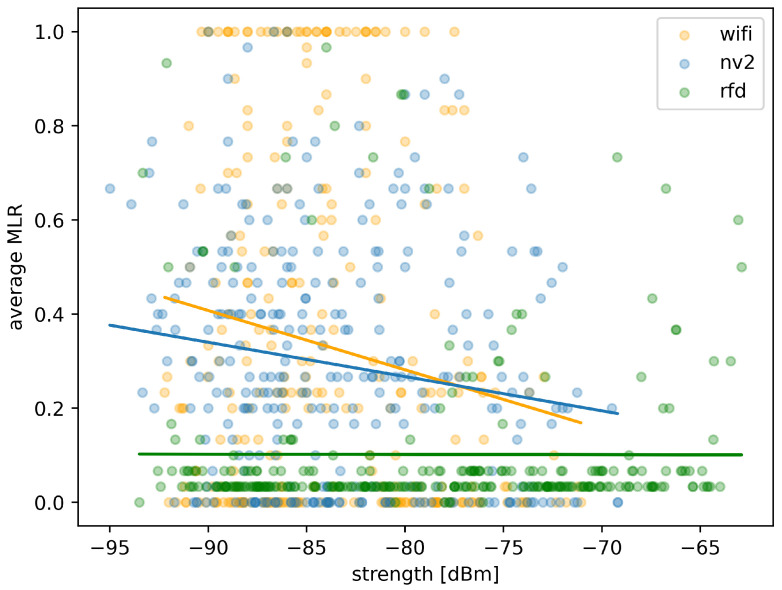
Scatter plot showing correlation between MLR and received signal strength level.

**Figure 16 sensors-22-00655-f016:**
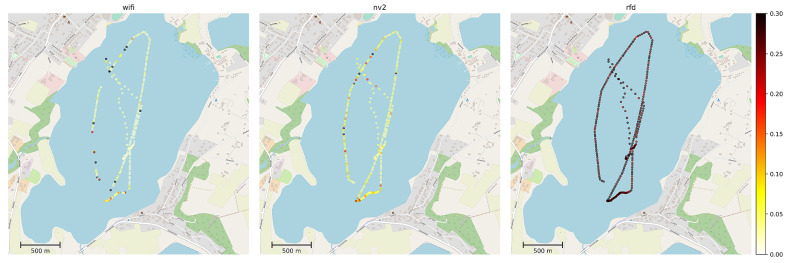
Round trip time measurements for MAVLink message exchange over Wi-Fi, NV2 and RFD links.

**Figure 17 sensors-22-00655-f017:**
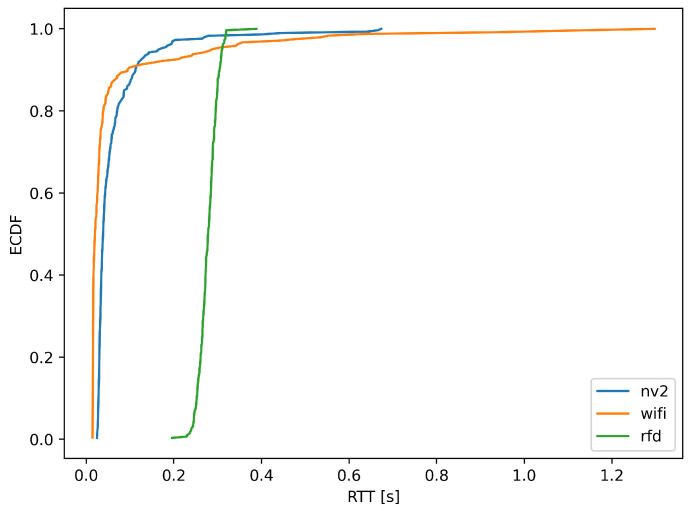
ECDF plot of message round trip time results.

**Figure 18 sensors-22-00655-f018:**
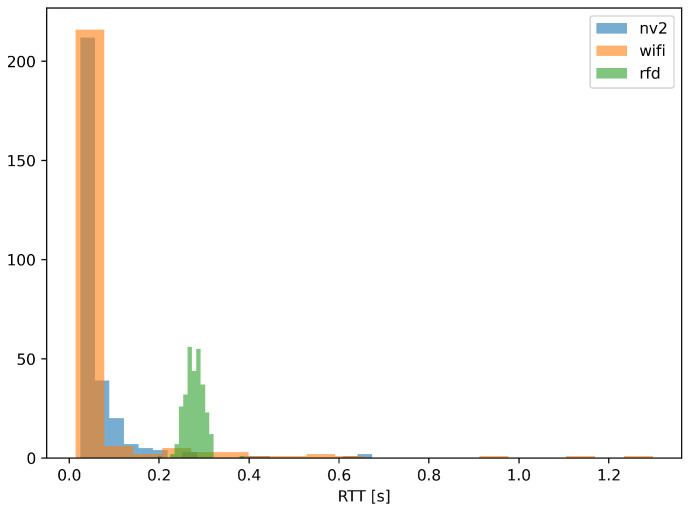
Histogram of message round trip time results.

**Figure 19 sensors-22-00655-f019:**
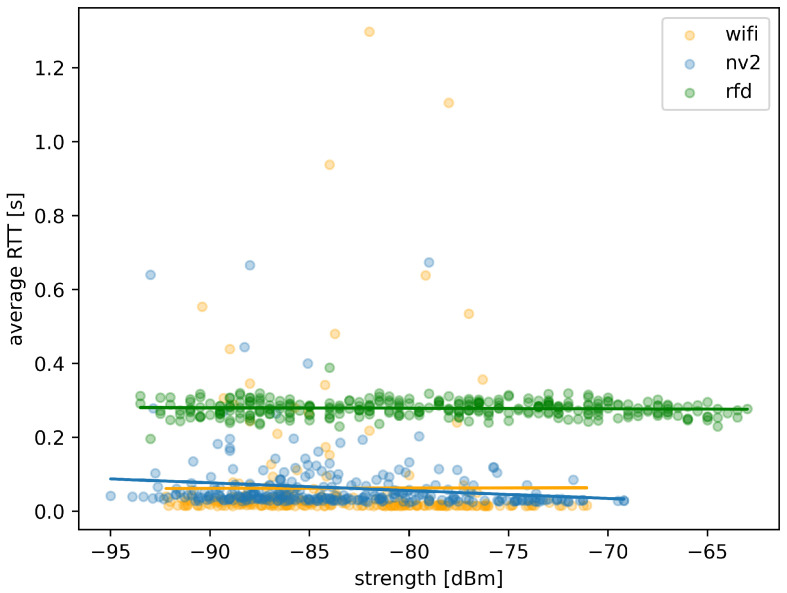
Scatter plot showing correlation between RTT and received signal strength level.

**Figure 20 sensors-22-00655-f020:**
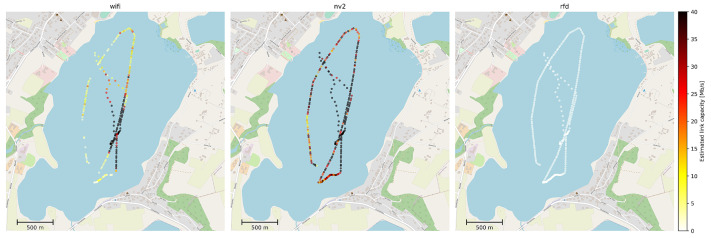
Assessment of the free communication bandwidth available with different transmission technologies.

**Table 1 sensors-22-00655-t001:** A summary overview of experiment results.

Technology	Maximum Recorded Communication Distance (m)	MAVLink Message Loss Ratio	MAVLink Message Round Trip Time (s)
NV2 5 GHz	2263	0.29	0.06
Wi-Fi 5 GHz	2264	0.36	0.06
Si1000 (RFD) 868 MHz	2264	0.09	0.28
Si1000 433 MHz	890	0.90	0.23

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
