# Peer review of "Wireless Local Area Network Technologies as Communication Solutions for Unmanned Surface Vehicles"

_sensors, 2022, doi:10.3390/s22020655_

Round 1

Reviewer 1 Report

The paper is quite well written such the reviewer would like to suggest its acceptance. There is only one minor concern of the reviewer is about how surrounding communication traffic (using the same band) will affect the performance of the designed network. It seems to the reviewer that the authors employ unlicensed bands for the dedicated network. So how interference and congestion of other systems utilizing the same bands would be an issue.

Author Response

Reviewer 1

The paper is quite well written such the reviewer would like to suggest its acceptance. There is only one minor concern of the reviewer is about how surrounding communication traffic (using the same band) will affect the performance of the designed network. It seems to the reviewer that the authors employ unlicensed bands for the dedicated network. So how interference and congestion of other systems utilizing the same bands would be an issue.

Both commonly used Sik radios and WLAN systems utilize IMS unlicensed frequency bands. Thus, all of them have to face problems of interference and congestion. However, as it was also discussed in the paper, WLAN networks offer much more bandwidth, so the impact of surrounding systems using the same frequency and other traffic sent in-band would be more significant in Sik radio links. As WLAN systems have a number of mechanisms (i.e.  prioritization, collision avoidance, channel width adjustment) supporting a coexistence of similar systems it would be much more reliable. As these aspects of IP network usage in SUV control systems are important and wide topics, in the first paper covering these fields we decided to verify if transmission parameters measured using application end-to-end point of view, are comparable with widely used Sik interfaces. Such results allow us to establish a baseline which will be used as a reference to study the impact of the interference sources on the communication.

To highlight our approach, an additional clarification has been introduced in the “Test system design” section (page 12).

Reviewer 2 Report

This paper aims to apply the popular Internet Protocol(MAVLink protocol) for surface unmanned vehicles. However, the main contribution of this paper is unclear to the reviewer. If the authors just use the predefined protocol without any novel approaches, the originality of this work is low. The detailed comments are as follows.

1) The authors need to summarize their main contributions in the introduction part.
2) The authors need to analyze other similar approaches for surface unmanned vehicles and compare with the implemented system.
3) Several figures are not clear. Please use the high resolution figures.
4) Even with the protocol's content, only applying it to a specific system may not be enough for a journal publication. Please describe the technical challenges and the authors' own solutions.

Author Response

This paper aims to apply the popular Internet Protocol (MAVLink protocol) for surface unmanned vehicles. However, the main contribution of this paper is unclear to the reviewer. If the authors just use the predefined protocol without any novel approaches, the originality of this work is low. The detailed comments are as follows.

1) The authors need to summarize their main contributions in the introduction part.

In the revised submission, our contribution has been clarified to highlight new aspects and technical challenges covered in the paper.  

2) The authors need to analyze other similar approaches for surface unmanned vehicles and compare with the implemented system.

As one of the main goals of the paper, we have chosen a comparison of narrowband Sik radio systems operating in Sub-1GHz frequencies (commonly used for external USV communication tasks, for example in case of HydroDron platform), with broadband technologies commonly employed for general IP communication, but as yet not USV-specific. Selected broadband technologies belong to similar price range as Sik solutions, and each of them employs a radically different medium access control mechanism. In other words, the paper concentrates on a comparison of currently employed, dedicated technologies (which can be seen as state-of-the-art) with general-purpose solutions potentially offering unique advantages but requiring specific configuration of their parameters (which were deployed and configured by authors as elements of the test system).

The results indicate that the 433 MHz Sik radio is significantly less performant than the 868 MHz Sik and WLAN systems (Table I). Of the remaining technologies, each offers unique advantages (as presented in plots and descriptions), with WLAN solutions proposed and deployed by authors proven to be capable of efficiently operating in roles previously occupied by narrowband solutions.

Following your remaining remarks, we have modified the introduction to state the focus of the paper clearly including the comparison described above.

3) Several figures are not clear. Please use the high resolution figures.

Thank you for pointing out this error - we have prepared the figures in higher resolution.

4) Even with the protocol's content, only applying it to a specific system may not be enough for a journal publication. Please describe the technical challenges and the authors' own solutions.

Obtaining measurement data analyzed in the paper required overcoming a number of challenges by designing and implementing relatively complex technical solutions. They are not described in high detail, as the main topic of the paper is a comparison and validation of a representative set of communication technologies, not the construction of the test system itself (and we felt that too many such details would cause the paper to lose its focus). Additionally, the effective use of MAVLink over IP transmitted by 4 devices belonging to two radically different technology groups requires appropriate configuration of a number of devices and mechanisms located in several layers of the ISO-OSI model. With that in mind, we would like to specify the following as our technical achievements:

  1. A suitability of MAVLink over IP communication for external UV communication tasks verification in a field-deployment scenario.
  2. A suitability of WLAN technologies and their inexpensive COTS implementations for MAVLink over IP verification in a field-deployment scenario.
  3. A comparison of broadband communication technologies belonging to the most popular classes of WLAN solutions (TDMA and contention-based) deployed in support for UV-specific protocol, with narrowband technologies commonly deployed for such tasks.
  4. Design and implementation of a hardware/software test system allowing easy comparison (by employing a non-interfering parallel testing technique) of employed technologies and its deployment both in laboratory conditions and on a professional-grade surface UAV during real-world deployment. The system employs (and allows comparison of results) both narrowband and WLAN technologies, despite their different operation principles and provided physical interfaces.
  5. Results establishing a baseline for further research concerning UV external communications due to minimization of the impact of external interference sources during field-grade tests. Such reference results facilitate further studies (especially interpretation of results) in areas where such an interference is present. Obtained measurement dataset includes extensive information regarding communication devices state and allows diverse analysis of traffic Quality of Service parameters. For the purpose of the analysis provided in the paper, an application level end-to-end Round-Trip Time and Message Delivery Ratio of MAVLink communication has been analyzed.

Keeping in mind your previous remarks, we added the above information in the revised introduction.

Reviewer 3 Report

This paper provides the results of conducted experiments to verify the suitability of Wireless Local Area Network (WLAN) technologies for the purpose of communication of surface Unmanned Vehicles (UVs). The topic addressed in this paper is potentially relevant to the readers of the journal. The title, abstract and introduction are appropriate. The overall organization of this paper is fine. However, there are some issues that need to be resolved.

  • The terms should be expanded at first place when they are introduced in the paper, such as Micro Air Vehicle Link (MAVLink) protocol should be expanded in the Introduction page 2 - line 84.
  • Figure 1 only shows two-way arrows among elements and provide the abstract level of information. It would be good to make Figure 1 more readable and provide more details, such as what data/information shared among these elements and how sequence flows (e.g., inputs and outputs).
  • It would be good to use italic or bold text when explaining a specific system to quickly grasp the reader, for instance, “The propulsion control system is responsible …” in Section 2
  • Page 4 - line 155 “Should not require human input…” -> “fully autonomous vehicles have not required ...”
  • Sometimes lidar is written as LiDAR. It should be consistent throughout the paper.
  • There are several English mistakes like “can protected even”, “either are or in our opinion”, “The technologies do no”, etc.
  • Page 9 – line 352 “as defined by European Telecommunications Standards Institute (ETSI)” the standard reference should be provided.
  • Page 16 - line 664 TDMA term suddenly introduced, however, when Time Division Multiple Access is used its abbreviation (TDMA) should be provided at page 11.
  • In Section 5 the block diagram of HydroDron system should be provided to increase the understandability of the readers.
  • It would be good if parts in Figure 6 will be highlighted like antenna, camera, LiDAR etc.

Author Response

Reviewer 3

This paper provides the results of conducted experiments to verify the suitability of Wireless Local Area Network (WLAN) technologies for the purpose of communication of surface Unmanned Vehicles (UVs). The topic addressed in this paper is potentially relevant to the readers of the journal. The title, abstract and introduction are appropriate. The overall organization of this paper is fine. However, there are some issues that need to be resolved.

Figure 1 only shows two-way arrows among elements and provide the abstract level of information. It would be good to make Figure 1 more readable and provide more details, such as what data/information shared among these elements and how sequence flows (e.g., inputs and outputs).

Thank you very much for your suggestions regarding improving editorial layout of the paper and pointing out a number of errors which escaped our attention – we have corrected the mistakes and highlighted subsystem names in the description of Fig. 1. However, we found it difficult to include additional specifics regarding interactions between subsystems in this description. The general architecture diagram has been prepared as a generalization which remains valid for as wide range of UV types as possible, which is a difficult task given their variety. A number of possible standards concerning physical connections, electrical signals, communication protocols, logical message formats and their flow sequences makes anything more detailed specific to an arbitrary and limited group of UVs, as use of specific solutions is a design decision of an engineer. We also need to keep in mind that the paper is dedicated to external communication mechanisms (a single element in the figure) with some references to internal communications and we are unable to include an extended analysis of complete set of UV subsystems without the paper losing its main focus. For similar reasons, we were unable to include a detailed schematic showing interconnection of Hydrodron systems (additionally it is a proprietary construction and many details are confidential).  However, in section 5 we have provided an annotated view of the specific UV employed in the experiments, illustrating the placement of its subsystems and payload.

Detailed remarks – group I:

  • It would be good to use italic or bold text when explaining a specific system to quickly grasp the reader, for instance, “The propulsion control system is responsible …” in Section 2
  • The terms should be expanded at first place when they are introduced in the paper, such as Micro Air Vehicle Link (MAVLink) protocol should be expanded in the Introduction page 2 - line 84.
  • Sometimes lidar is written as LiDAR. It should be consistent throughout the paper.
  • Page 9 – line 352 “as defined by European Telecommunications Standards Institute (ETSI)” the standard reference should be provided.
  • Page 16 - line 664 TDMA term suddenly introduced, however, when Time Division Multiple Access is used its abbreviation (TDMA) should be provided at page 11.
  • Page 4 - line 155 “Should not require human input…” -> “fully autonomous vehicles have not required ...”
  • There are several English mistakes like “can protected even”, “either are or in our opinion”, “The technologies do no”, etc.

Thank you for these remarks. All of them have been addressed and corrected in the newest version of the submission.

Detailed remarks – group II:

  • In Section 5 the block diagram of HydroDron system should be provided to increase the understandability of the readers.
  • It would be good if parts in Figure 6 will be highlighted like antenna, camera, LiDAR etc.

As explained in the first response, we were unable to include a block diagram of HydroDron showing interconnection of internal systems (additionally it is a proprietary construction and many details are confidential).  However, in section 5 we have provided an annotated view of the specific UV employed in the experiments, illustrating the placement of its subsystems and payload – Fig. 7.

Round 2

Reviewer 2 Report

You have done this revision well. The paper is acceptable.